# Green Synthesis of Phosphorescent Carbon Dots for Anticounterfeiting and Information Encryption

**DOI:** 10.3390/s22082944

**Published:** 2022-04-12

**Authors:** Mingming Cheng, Lei Cao, Hanzhou Guo, Wenfei Dong, Li Li

**Affiliations:** 1School of Biomedical Engineering (Suzhou), Division of Life Sciences and Medicine, University of Science and Technology of China, Hefei 230026, China; cmm13862591726@mail.ustc.edu.cn (M.C.); cl1221@mail.ustc.edu.cn (L.C.); wenfeidong@sibet.ac.cn (W.D.); 2Changchun Guoke Medical Engineer and Technology Development Co., Ltd., Changchun 130033, China; guohanzhou1@163.com; 3CAS Key Laboratory of Biomedical Diagnostics, Suzhou Institute of Biomedical Engineering and Technology, Chinese Academy of Science (CAS), Suzhou 215163, China

**Keywords:** room-temperature phosphorescent, carbon dots, anti-counterfeiting, information protection

## Abstract

Room-temperature phosphorescent (RTP) carbon dots (CDs) have promising applications in bioimaging, anticounterfeiting, and information encryption owing to their long lifetimes and wide Stokes shifts. Numerous researchers are interested in developing highly bright RTP CDs using environmentally friendly and safe synthesis processes (e.g., natural raw materials and zero-pollution production pathways). In this study, we successfully synthesized RTP CDs using a hydrothermal process employing natural vitamins as a raw material, ethylenediamine as a passivator, and boric acid as a phosphorescent enhancer, which is referred to as phosphorescent CD (PCD). The PCDs exhibit both bright blue fluorescence emission and green RTP emission, with a phosphorescence lifetime as long as 293 ms and an excellent green afterglow visible to the naked eye for up to 7.0 s. The total quantum yield is 12.69%. The phosphorescence quantum yield (PQY) is up to 5.15%. Based on the RTP performance, PCDs have been successfully employed for anticounterfeiting and information protection applications. The results of this study provide a green strategy for the scalable synthesis of RTP materials, which is a practical method for the fabrication of RTP materials with high efficiency and long afterglow lifetimes.

## 1. Introduction

Room-temperature phosphorescent (RTP) materials have attracted increasing attention because of their enormous potential in optoelectronics [1,2], ensing [3,4,5], imaging [6,7], and safe encryption [8,9,10,11,12]. Metal-doped inorganic [13,14] and organometallic compounds [15,16,17] are now the most effective RTP materials. However, these materials, which are frequently composed of rare metals, are expensive, cytotoxic, rigid, and unstable. Pure organic RTP materials [18,19,20], as an alternative, are less effective because of their low intersystem crossing (ISC) and rapid deactivation rate. Thus, it is very desirable to develop new metal-free RTP materials with low toxicity, environmental protection, long life, and low cost.

In recent years, novel zero-dimensional carbon nanomaterials, known as carbon dots (CDs), have attracted significant interest because of their low toxicity, biocompatibility, ease of manufacture, and remarkable optical properties [20,21]. However, generating phosphorescent CDs at room temperature is very challenging because of the instability of excited triplet species, oxygen-induced phosphorescence quenching, and inefficient intersystem crossover (ISC). There are two distinct strategies for achieving room-temperature phosphorescence in CDs. One method is to introduce heteroatoms, which facilitate efficient spin–orbit coupling and result in a low singlet–triplet splitting energy [22,23,24]. Typically, nitrogen–phosphorus co-doped CDs are synthesized by microwave irradiation using triethanolamine as the carbon source and phosphoric acid as the dopant [25]. Another possibility is to embed CDs in various matrices, including polyvinyl alcohol [26,27], acrylamide [5,28], urea [1], trisodium citrate [2], aluminum sulfate [29], zeolite, silica [30,31,32], melamine [33,34], and boric acid (BA) [35,36]. Recently, by embedding CDs in a silica matrix, a 5.72-s long-life RTP material was developed. During this process, the adequate three-dimensional nano-space encompassed by the Si-O tetrahedra acts as a spatial constraint, substantially suppressing intramolecular vibrations, the development of stable covalent and hydrogen bonds between the rigid structure, and CDs successfully stabilizing their triplet excited states.

At present, many valuable methods for synthesizing phosphorescent CDs have been developed [37,38,39]. The main commonly used methods are one-step synthesis and two-step synthesis. The one-step method also includes the hydrothermal method, microwave method, heat treatment, etc. The reaction precursors such as different carbon sources, organic materials, heteroatoms, etc. are mixed, and then, the reaction temperature and time are selected to obtain the reaction products in one step. Although this method is simple and fast, the process of in situ growth of carbon dots on the substrate is highly uncontrollable and usually requires a strong driving force to achieve. Therefore, this approach is not generally considered [40,41,42,43]. However, the two-step approach is synthesizing carbon dots first and then embedding them into different substrates such as nanoporous materials, zeolites, polyvinyl alcohol, boric acid, etc. [40,41,42,43,44,45,46]. This method is more controllable and easier to implement than the one-step method [47]. Therefore, in this study, the two-step method was used to synthesize phosphorescent carbon dots [48].

It is well-known that the energy gap between the single and triplet states can be reduced by the boron atom, because B is an electron-absorbing atom. When BA is introduced into the CDs, it favors the ISC from S1 to T1, which enhances the phosphorescence emission of the triplet exciton [20,21]. In brief, when BA is used as a matrix, the generated glassy state prevents the CD’s triplet excitons from being consumed nonradiative. Additionally, as an electron-attracting atom, boron possesses an empty p orbital that may attract transitions to create a p-conjugate system, thereby lowering the S1–T1 energy gap and increasing the rate of ISC [49,50,51]. More importantly, the covalent connections produced between the B and C atoms stabilize the overall system [36,37], thus facilitating RTP emission. Owing to these characteristics, BA is regarded as a promising matrix for the universal fabrication of CD-based RTP materials. However, very few studies have been conducted on the use of BA as a phosphorescent emission substrate.

Herein, by a two-step approach, we synthesize carbon dots first and then embed them into boric acid substrates (Figure 1). This method is more controllable and easier to implement than the one-step method [52]. Besides, we prepared phosphorescent CDs (PCDs) with vitamin B1, a natural vitamin, as the main raw material. The preparation process is nontoxic, harmless, and environment-friendly. The preparation RTP materials have low toxicity, environmental protection, and low costs. The prepared PCDs show excellent photoluminescence (PL) and phosphorescence. The total quantum yield is 12.69%. The phosphorescence quantum yield (PQY) is as high as 5.15%, the RTP life is 293 milliseconds, and the visible afterglow lasts more than 7 s. Owing to the excellent RTP performance, the PCDs have been successfully employed for anticounterfeiting and information protection applications.

## 2. Materials and Methods

### 2.1. Chemicals and Materials

Thiamine hydrochloride (Vitamin B1), ethylenediamine (EDA), and boric acid (BA) were purchased from Titan Scientific (Shanghai, China). Ultrapure water was used throughout the whole experiment.

### 2.2. Instrumentation

High-resolution transmission electron microscopy (HR-TEM) pictures were collected at 100 kV using a TECNAI G2 microscope (Thermo Fisher, Waltham, MA, USA). The image of scanning electron micrographs (SEM) was measured by a XL-30ESEM FEG scanning electron microscope (FEI, Hillsboro, OR, USA). FTIR spectra were conducted with a VERTEX 70 FT-IR spectrometer (Bruker, Bremen, Germany). The X-ray photoelectron spectroscopy (XPS) spectra were acquired using an ESCALAB 250Xi spectrometer (Thermo Fisher, Waltham, MA, USA). A Rigaku Minister apparatus was used to generate the X-ray diffraction (XRD) patterns (Tokyo, Japan). The UV absorption spectra were performed via a Hitachi UV2450 spectrophotometer (Tokyo, Japan). The fluorescence spectra were obtained using an F97Pro FL spectrophotometer coupled with a 1.0-cm quartz cell (Lengguang Technology, Shanghai, China). Additionally, the fluorescent, phosphorescent lifetime, and emission spectrum were analyzed at room temperature using an FLS-1000 fluorescence spectrophotometer (Edinburgh, UK).

### 2.3. Synthesis of VB1-CDs

In general, 0.5 g of vitamin B1 was dissolved in 10 mL of ultrapure water initially, and then, 150 µL of EDA was dropped into the solution and ultrasonically dispersed well. Subsequently, the solution was transferred to a 20-mL poly(tetrafluoroethylene)-lined autoclave and reacted at 180 °C for 8 h in a drying oven. After the reaction, the centrifuge (8000 rpm, 10 min) removed the pellets and prepared them for usage.

### 2.4. Synthesis of PCDs

Basically, 50 µL, 100 µL, 500 µL, and 1000 µL of VB1-CD solutions were, respectively, added into small beakers, filled with 20-mL ultrapure water, and stirred evenly. Then, 3 g of boric acid was added into each beaker, covered with tin foil, and placed in a drying oven for reaction at 180 °C for 5 h. After the reaction, the materials were ground into powder; thus, PCD50, PCD100, PCD500, and PCD1000 were obtained, respectively.

### 2.5. Fabrication of LEDs

The central LED chips, which emit 395-nm UV and 460-nm blue light, were purchased from Shenzhen Prospect Technology Co (Shenzhen, China). The LEDs all operated at a voltage of 3.0 V. PCD100 powder was mixed with epoxy resin AB glue and then placed at the center of the LED chips. After drying in a 100 °C oven for one hour, LED beads were obtained.

## 3. Results and Discussion

### 3.1. Characterization and Optical Properties of VB1-CDs

Transmission electron microscopy (TEM) was performed to investigate the morphology of the VB1-CDs (Appendix A). The FTIR spectrum and XPS characterizations of VB1-CDs are shown in Appendix A. The measured XPS spectrum of VB1-CDs in Appendix A clearly shows four peaks at 285.14, 400.16, 531.38, and 163.56 eV for C 1 s, N 1 s, O 1 s, and S 2 p, respectively. The elemental contents of the VB1-CDs were 73.96% C, 12.11% O, 11.84% N, and 2.46% S, respectively. A series of analyses such as FTIR and XPS on the VB1-CDs show that there are C=N and C=O in the materials, etc. C=N and C=O promote fluorescence emission. Steady-state spectroscopic investigations were conducted to elucidate the photophysical behavior of VB1-CDs. Figure 2 depicts the UV–Vis absorption, excitation, and fluorescence spectra of VB1-CDs. The UV absorption (Figure 2a) results in the formation of two humps at 271 and 320 nm, which are ascribed to the n–π* transition of C–O and C=O or C–N functional groups present at the margins of the VB1-CDs. The π→π* leap of the conjugated C=C unit in the carbon nucleus produces a strong absorption near 228 nm. The presence of these groups beneficially influences the fluorescence properties of the CDs [38,39]. The VB1-CDs exhibit an emission of 431 nm with excitation at 365 nm, with no noticeable absorption peak at 400 nm (Figure 2a). As shown in the inset of Figure 2b, the fluorescence wavelength is red-shifted from 423 to 525 nm, with a consistent increase in the excitation wavelength from 325 to 465 nm, demonstrating apparent excitation-dependent behavior. The highest excitation and emission wavelengths are 385 and 438 nm, respectively. The absolute quantum yield of VB1-CDs is 9.49%, and Figure 2c shows the fluorescence lifetime decay with a lifetime of 4.86 ns.

### 3.2. Characterization of PCDs

Taking PCD100 as an example, transmission electron microscopy (TEM) was performed to investigate the morphology and particle distributions of the PCDs, which demonstrates that the PCDs are evenly distributed as spherical shapes with an average size of 2.3 nm (Figure 3a). The size distribution of PCD100 is shown in Appendix A, and the results indicate that the diameters of PCD100 range from 1.03~3.98 nm. Additionally, PCD100 has a lattice fringe spacing of 0.21 nm, which is consistent with the categorization of graphite carbon. Appendix A shows the scanning electron microscope (SEM) images of the powders of PCD100 [25,40].

To further investigate the functional groups and chemical composition of the PCDs, Fourier-transform infrared spectroscopy (FTIR) and X-ray photoelectron spectroscopy (XPS) techniques were applied. PCD100 exhibits a wide FTIR absorption band between 2500 and 3500 cm^−1^, as depicted in Figure 3b, indicating the presence of O-H (3250 cm^−1^) on the exterior. In addition, the peaks at 1484 cm^−1^ are ascribed to the stretching vibrations of C–N. Stretching vibrations of hydrogen-bonded C–O–B (1232 cm^−1^) and C–B (911 cm^−1^) are observed, demonstrating the presence of B doping in the PCDs [35].

The presence of O, C, and B are visible in the full-scan XPS spectrum of PCD100 (Figure 3c), with atomic percentages of 38.3%, 32.18%, and 27.45%, respectively. The characteristic peaks for O1, C1, and B1s are 531.08 eV, 285.08 eV, and 193.08 eV, respectively. The XPS spectra of C1s at high resolution (HR) display three components: C–C/C=C (284.7 eV), C–O (286.3 eV), and C=O=O (288.7 eV) (Figure 3d). The three peaks in the HR B1s XPS spectra at 192.6 eV, 193.6 eV, and 194.4 eV are assigned to BCO2, B2O3, and B-O, respectively (Figure 3e) [41,42,43]. The XPS results are compatible with the FTIR findings. X-ray diffraction (XRD) analysis of PCD100 shows characteristic B2O3 peaks at 14.5°, 27.4°, 30.2°, and 40.0° (Figure 3f).

### 3.3. Optical Properties of PCDs

The photophysical characteristics of PCD100 were investigated thoroughly. The UV–Vis absorption spectra of the PCD100 aqueous solution exhibit three peaks at 220, 260, and 322 nm (Figure 4a). The first one is assigned to the π–π* transformation of C=C, while the last two emerge from the n–π* transformation of C=O [41]. The PL spectra of the PCD100 aqueous solution exhibit excitation-dependent characteristics, with the greatest emission occurring at 430 nm under 365-nm excitation (Figure 4b). However, no afterglow is observed in the PCD100 solution, which is attributed to molecular rotation, vibration, and collisional triplet relaxation. The emission of solid PCD100 fluorescence (Figure 4c) and phosphorescence (Figure 4d) are excitation-dependent, with the highest emissions at 496 nm and 570 nm, respectively. The fluorescence emission of solid PCD100 moves gradually from 431 to 507 nm as the excitation wavelength increases (Figure 4c).

The maximum PQY of PCD100 is as high as 5.15%, which is why PCD100 produces bright blue light when exposed to a UV lamp (254 nm) and still leaves a light-green afterglow visible to the naked eye when the lamp is turned off, which lasts for about 3 s (Figure 5a). Furthermore, when excited at 365 nm, PCD100 emits blue fluorescence, and ultralong-lived yellow-green RTP is identified up to 7 s after the excitation source is removed (Figure 5b). As shown in Figure 5c, the degradation of PCD100 can be modeled using a triexponential function, and the average lifetime is determined to be 293 ms at 365-nm excitation. The fluorescence lifetime decay is shown in Figure 5d, and the lifetime is 5.50 ns. These findings suggest that the presence of several triple-excited states may be responsible for the excitation-dependent properties of PCD100. The elimination of the afterglow in the solution implies that the performance of RTP may be stabilized in the solid state. In order to understand our research results more clearly, we compared them with previous studies in terms of RTP lifetime and PQY, and the results are shown in Appendix A.

To elucidate the nature of the RTP, a series of controlled studies were conducted. No RTP is identified when only VB1-CD or BA is used as a precursor. Under identical conditions, 3 g of BA interacted with various masses (50, 500, and 1000 µL) of VB1-CD to produce the PCDs, named PCD-50, PCD-500, and PCD-1000, respectively. PCD-50, PCD-500, and PCD-1000 have absorption spectra identical to those of BD50, exhibiting three characteristic peaks at 224, 252, and 323 nm (Appendix A). The PL spectra of PCD-50 (Appendix A), PCD-500 (Appendix A), and PCD-1000 (Appendix A) aqueous solution exhibited excitation-dependent characteristics, with the greatest emission occurring at 432, 474, and 473 nm under 365, 405, and 405-nm excitation, respectively. PCD-50, PCD-500, and PCD-1000 solids display excitation-dependent PL characteristics, with emission peaks at 474, 488, and 500 nm, respectively, when excited at 420 nm (Appendix A, respectively). Appendix A shows the phosphorescence spectra of PCD-50, PCD-500, and PCD-1000, with the greatest emission occurring at 521, 539, and 551 nm under 340, 380, and 380-nm excitation, respectively. The average fluorescence lifetime of PCD-50, PCD-500, and PCD-1000 is 5.33, 5.51, and 5.41 ns, respectively (Appendix A; Appendix A). When activated by a UV lamp at 254 nm, PCD50 emits light-blue light, PCD500 emits light-yellow light, and PCD1000 emits orange light (Appendix A). When the UV light at 254 nm is shut off, PCD50 displays a very weak green RTP that lasts 3 s, whereas PCD500 displays a very weak pale yellow RTP that lasts only 2 s. However, when the 365-nm UV light is shut off, PCD50 displays a strong green RTP that is visible to the human eye for 6 s, PCD500 displays a green RTP that is visible to the naked eye for 5 s, and PCD1000 displays a light-green RTP for 3 s (Appendix A). The total quantum yield of PCD-50, PCD-500, and PCD-1000 are 12.08%, 7.18%, and 4.65%, respectively. By calculating the spectral integrated area, the PQYs of PCD-50, PCD-500, and PCD-1000 are 4.25%, 4.00%, and 2.82%, respectively [53,54]. The average lifetime (Appendix A and Appendix A) of PCD-50, PCD-500, and PCD-1000 reduces from 292, 207, and 152 ms, respectively. Based on the findings mentioned above, the fluorescence and phosphorescence properties of PCD100 are the best.

The XRD patterns of PCD-50, PCD-500, and PCD-1000 exhibit distinctive peaks comparable to those of PCD-100 (Appendix A). XPS analysis was used to identify the chemical compositions of PCD-50, PCD-500, and PCD-1000 (Appendix A). The B contents of PCD-50, PCD-500, and PCD-1000 are 32.93%, 34.52%, and 27.35%, respectively (Appendix A). The S contents of PCD-50, PCD-500, and PCD-1000 are 0.81%, 1.51%, and 0.52%, respectively (Appendix A). These findings suggest that the B–C covalent connections produced between VB1-CDs and BA may enhance the RTP characteristics of the PCDs. The FTIR spectra of PCD-50, PCD-500, and PCD-1000, which are comparable to those of PCD100, are shown in Appendix A.

Taking PCD50 and PCD100 as examples, we measured their fluorescence and phosphorescence spectral data on day 1 and day 7, respectively, and compared the data to verify the stability of the material (Appendix A). The results show that the spectral values of PCDs are slightly decreased, but the changes are not very large, so we can assume that the fluorescence and phosphorescence properties of phosphorescent carbon dots are more stable.

Considering the prior discussion, we propose a plausible mechanism for the phosphorescence of the PCDs. First, the C=O=O bonds in PCDs can induce RTP. Second, BA is necessary for synthesizing RTP, which may generate C–B bonds when combined with VB1-CD. The covalent bonds and nanoconfined space of BA may prevent the extinguishing of excited triplet excitons, resulting in the facilitation of RTP emission. Based on the PL spectrum and the phosphorescence emission spectrum of PCD50 and PCD100. According to this formula, EST = h/k × C/λ = 1240/λ, the energy gaps (ΔE_ST_) between the lowest single (S1) and triplet (T1) states were calculated to be equal to 0.42 eV and 0.43 eV, which are small values, allowing an effective ISC process to occur (Appendix A) [35,36,54].

### 3.4. Application in LED, Anticounterfeiting, and Information Security

The high-efficiency blue-green fluorescence–phosphorescence emission of PCDs, together with their low cost and environmental friendliness, make them attractive candidates for use in high-performance single-component white light-emitting diodes (WLEDs). The LED chips operate at a voltage of 3 V and a current of 150 mA. The UV-pumped WLED was fabricated by mixing PCD100 powder with epoxy resin AB glue and then placed at the center of the 395-nm UV-LED chips and 460-nm blue LED chips. At a voltage of 3.0 V, the 395-nm UV LED produces efficient white emission. In addition, the LED exhibits green phosphorescence when the voltage is withdrawn (Appendix A). As shown in Figure 6b, highly efficient single-component WLEDs based on PCDs have electroluminescence (EL) spectrum in the range 380~750 nm. Two separate peaks are visible at 450 nm for blue fluorescence and 570 nm for green phosphorescence emission. The 395-nm UV LED generates white light with CIE color coordinates of (0.26, 0.30). The coordinates of the 460-nm blue LED are (0.17, 0.16) (Figure 6a), and the emission spectrum of the as-fabricated LEDs is shown in Figure 6b [42,43].

In addition, PCDs have significant potential in the anticounterfeiting and information security domains because of their higher URTP capacity. First, PCD100 was utilized as a model to illustrate its potential application as a smart material for security protection. Figure 7a displays a typical triple-modal switching encryption created by placing PCD100 powder into the molds. In daylight, the theme is white. When exposed to 365-nm light, the pattern fluoresces blue. A brilliant green RTP pattern emerges when the light is switched off. Furthermore, Figure 7b demonstrates the use of PCD powders for information encryption. The letters U and T, constructed of PCD100, are visible and display blue PL when excited at 365 nm. However, the letters S and C, made of non-phosphorescent material, produce yellow fluorescence. After deactivating the excitation, the green RTP emission “U T” could be distinguished from “USTC.”

Moreover, PCD100 can also be employed for anticounterfeiting purposes, because its phosphorescent emission is quenched by water. As shown in Figure 7c, the letters USTC are spelled out using PCD100; water is sprayed onto the second letter S and the fourth letter C. During the day, these four letters are hardly distinguishable. The initial letter U and the third letter T emit a blue glow when irradiated with 365-nm light. In comparison, the second letter S and the fourth letter C emit faint blue fluorescence. Only the first letter U and third letter T can be recognized as bright green phosphorescence when the UV lamp is switched off. These results indicate the practical use of PCDs in sophisticated anticounterfeiting and information protection applications.

## 4. Conclusions

We present a simple, environmentally friendly, and cost-effective method for generating RTP PCDs. PCD100 materials exhibit a high PQY (5.15%) value when excited at 365 nm. In addition, PCD100 has a long RTP lifetime of 293 ms with a visible afterglow duration of 7 s. Furthermore, PCDs have been effectively implemented for anticounterfeiting and data encryption. This study demonstrated a straightforward and highly successful approach for the synthesis of novel RTP luminophores utilizing commonly available and economic materials.

## Figures and Tables

**Figure 1 sensors-22-02944-f001:**
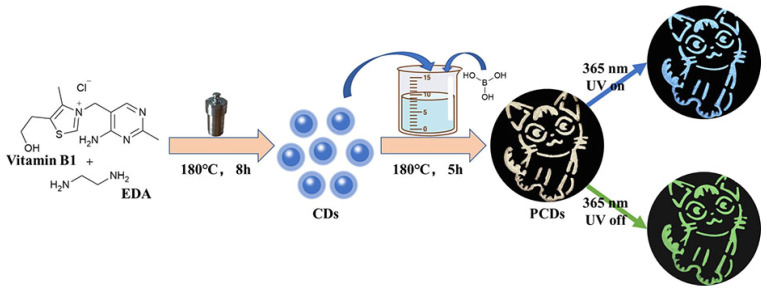
Schematic illustration of the synthesis process for the PCDs.

**Figure 2 sensors-22-02944-f002:**
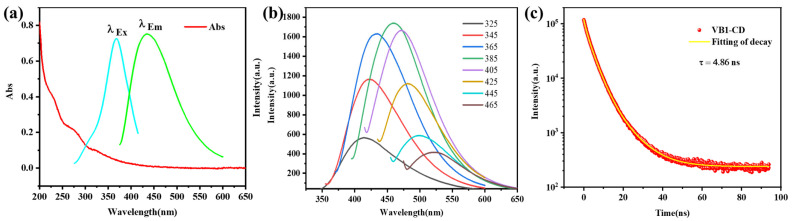
(**a**) UV–Vis absorption of VB1-CDs. (**b**) PL emission spectra of VB1-CDs aqueous solution. (**c**) Fluorescence lifetime decay of VB1-CDs.

**Figure 3 sensors-22-02944-f003:**
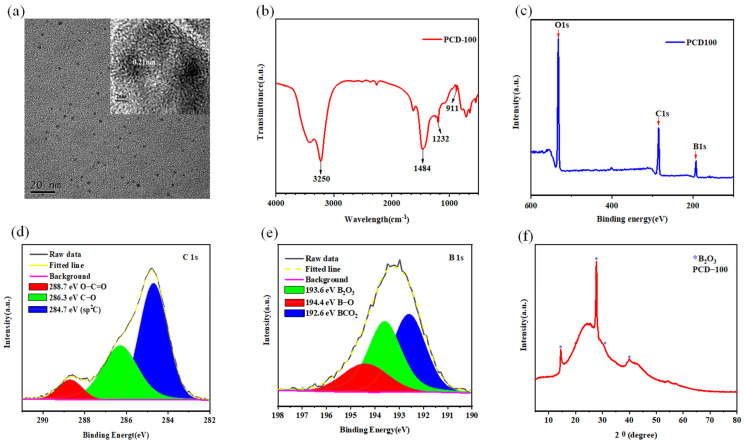
(**a**) TEM image of PCD100. (**b**) FTIR spectrum of PCD100. (**c**) Full-scan XPS spectrum of PCD100. (**d**) HR XPS C1s. (**e**) HR XPS B1s. (**f**) XRD pattern of PCD100.

**Figure 4 sensors-22-02944-f004:**
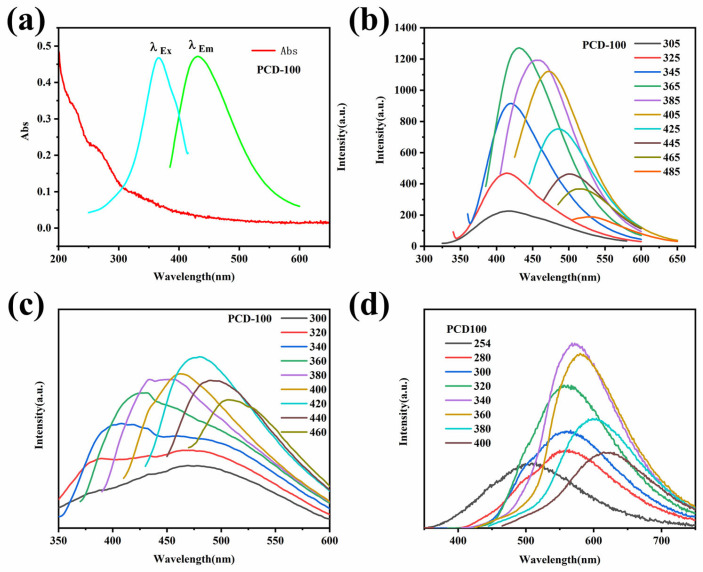
(**a**) UV–Vis absorption and PL excitation and emission spectra of PCD100 aqueous solution. (**b**) PL emission spectra of PCD100 aqueous solution. (**c**) PL emission spectra of PCD100 solid. (**d**) Phosphorescence emission spectra of PCD100 solid.

**Figure 5 sensors-22-02944-f005:**
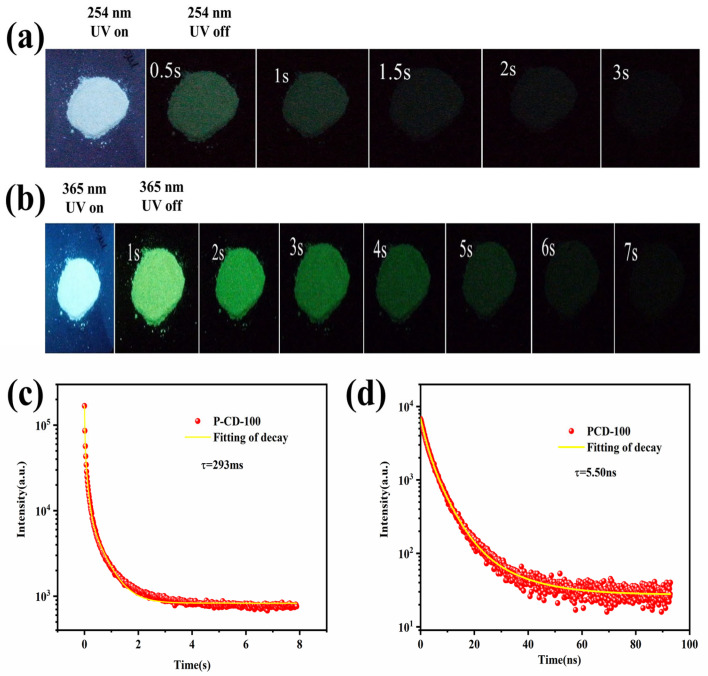
(**a**) Photographs of PCD100 with a 254-nm UV lamp on and off. (**b**) Photographs of PCD100 with a 365-nm UV lamp on and off. (**c**) RTP lifetime decay of PCD100 solid under the excitation of 365 nm. (**d**) Fluorescence lifetime decay of PCD100 solution.

**Figure 6 sensors-22-02944-f006:**
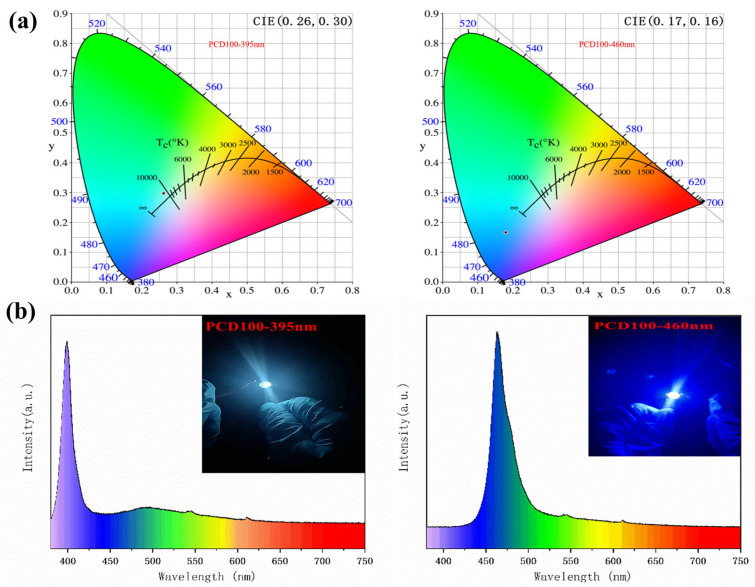
(**a**) CIE color coordinate of the WLED. (**b**) Emission spectrum of the WLED with UV.

**Figure 7 sensors-22-02944-f007:**
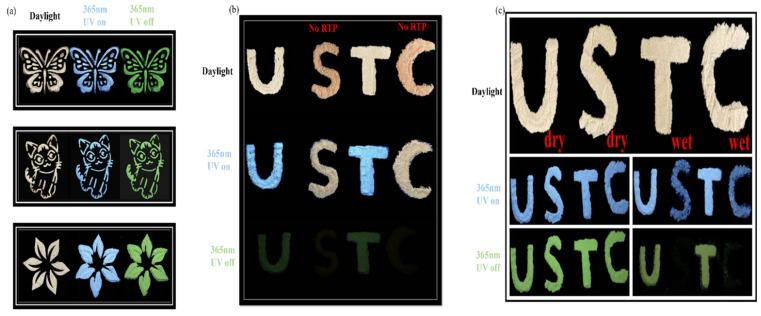
(**a**) Triple modal switching application of PCD100. (**b**) Anticounterfeiting application of PCD100. (**c**) Data encryption application of dry and wet PCD100.

## Data Availability

Not applicable.

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
