# Peer review of "Green Synthesis of Phosphorescent Carbon Dots for Anticounterfeiting and Information Encryption"

_sensors, 2022, doi:10.3390/s22082944_

Round 1

Reviewer 1 Report

In the present manuscript entitled Green synthesis of phosphorescent carbon dots for anti-counterfeiting and information encryption, the authors reported the preparation of room temperature phosphorescent (RTP) and explored their information encryption application.

The usage of boric acid as phosphorescent enhancer is well reported hence the novelty of this work is low. However, the authors generated significant amount of data that can add to the existing literature. Hence I suggest its acceptance after major revision.

The authors should add FTIR and XPS characterizations for VB1-CDs and compare them with PCDs-100. This will reveal the most important information regarding the binding interactions between VB1-CDs and BA in PCDs.

The data is not properly organized. Carefully check the figure numbers and appropriately cite them in the text (both in main manuscript and supporting information)

There are several mismatches of data in Figures and in text. For instance in line 126, the authors mentioned the wavelengths as 350 to 600 nm, however the actual values in Fig 2b are 325 nm to 465 nm. Please check the manuscript thoroughly and avoid all such mismatches.

Please group Fig S2 and S3; S4 and S5; Fig S7, S8 and S9. This will give a clear presentation the results.

In XPS elemental composition (Table S3), why the carbon content not increased from PCDs50 to PCDs1000 ? From PCDs50 to PCDs1000, the CDs content is increased; hence generally the Carbon content should also increase. However you got some random trend. How can you can you explain this. Please carefully check your results.

The explanation in lines 227 to 235 needs some supporting references.

Add the equation and appropriate citation for ΔEST calculation.

Reviewer 2 Report

I believe this manuscript has the merit to be published in the Sensors journal published by MDPI, but given the strict requirements of this journal, I recommend the authors address the following concerns via a major revision that would help to improve the manuscript. Please see the below comments:

  1. Why have the authors observed excitation-dependent emission for VB1-CDs? Please report the TEM data for VB1-CDs. 
  2. Are the shapes of PCDs really spherical? How do the authors make sure of that? Please provide the AFM data of PCDs and VB1-CDs?
  3. The authors should report the Raman data.
  4. I do not see any SEM images as Figure S1.
  5. Why is the fluorescence and phosphorescence of solid PCD100 excitation-dependent? Please explain.
  6. What are the origins of CDs' fluorescence and phosphorescence? 
  7. How did the authors measure the absolute quantum yield? Please explain the method.
  8. The authors should provide more device parameters such as turn-on voltage, EQE, IQE, etc.
  9. How did the authors measure the phosphorescence emission spectrum? Please include it in the instrumentation section.
  10. The authors should considerably extend the introduction section, including different types of methods used to synthesize CDs, explanation of choosing this method over others, merits/demerits of this/other methods?
  11. How did the authors purify these CDs in order to remove the bundles/aggregated particles and non-reactant precursors?
  12. How stable the fluorescence/phosphorescence was for a long time?
  13. There are some typos and minor grammatical mistakes in the manuscript. Please correct those carefully.
  14. The authors should consider improving the reference list by including more recent references.
  15. What is the novelty of this particular work? Please clearly explain it in the manuscript for the general readers.

Reviewer 3 Report

The review is attached

Round 2

Reviewer 1 Report

The authors addressed all my comments

Reviewer 2 Report

I believe the authors have made substantial changes to revise the manuscript. I recommend this manuscript be published in the Sensors journal by MDPI.